# Pulmonary Function Among COVID-19 Patients in Home Isolation Program

**DOI:** 10.3390/medsci13030088

**Published:** 2025-07-09

**Authors:** Narongkorn Saiphoklang, Pitchayapa Ruchiwit, Apichart Kanitsap, Pichaya Tantiyavarong, Pasitpon Vatcharavongvan, Srimuang Palungrit, Kanyada Leelasittikul, Apiwat Pugongchai, Orapan Poachanukoon

**Affiliations:** 1Department of Internal Medicine, Faculty of Medicine, Thammasat University, Pathum Thani 12120, Thailand; toon.pitchayapa@gmail.com (P.R.); apidoctor@hotmail.com (A.K.); pichaya_t@tu.ac.th (P.T.); 2Medical Diagnostics Unit, Thammasat University Hospital, Pathum Thani 12120, Thailand; lee.kanyada@gmail.com (K.L.); pu.apiwat@gmail.com (A.P.); 3Center of Excellence for Allergy, Asthma and Pulmonary Diseases, Thammasat University Hospital, Pathum Thani 12120, Thailand; orapanpoachanukoon@yahoo.com; 4Department of Clinical Epidemiology, Faculty of Medicine, Thammasat University, Pathum Thani 12120, Thailand; 5Department of Community Medicine and Family Medicine, Faculty of Medicine, Thammasat University, Pathum Thani 12120, Thailand; pasitpon@tu.ac.th (P.V.); srimuangpa@gmail.com (S.P.); 6Department of Pediatrics, Faculty of Medicine, Thammasat University, Pathum Thani 12120, Thailand

**Keywords:** COVID-19, home isolation program, pulmonary function, restrictive lung pattern, small airway disease

## Abstract

**Background**: Patients with mild coronavirus disease 2019 (COVID-19) are usually managed in an outpatient setting. Pulmonary functions in this setting have not been explored. This study aimed to determine abnormal lung functions in COVID-19 patients under a home isolation program. **Methods**: A prospective study was conducted in asymptomatic or mild COVID-19 patients with normal chest radiographs at two medical centers in Thailand. Spirometry data, including forced expiratory volume in 1 s (FEV1), forced vital capacity (FVC), peak expiratory flow (PEF), forced expiratory flow at 25–75% of FVC (FEF25–75), and bronchodilator responsiveness (BDR), were collected. Spirometry was performed after disease resolution at baseline and 3-month follow-up. Abnormal lung functions were classified into airway obstruction, restrictive defect, mixed defect, small airway disease, and BDR. **Results**: A total of 250 patients (58% female) were included. The mean age was 37.4 ± 15.2 years. Asymptomatic patients accounted for 7.6%. Common symptoms included fever (55.6%) and cough (60.0%). Abnormal lung functions were observed in 28.4% of patients, with a restrictive lung pattern (14.4%), airway obstruction (4.8%), mixed defect (0.4%), small airway disease (8.4%), and BDR (2.8%). Significant changes from baseline were noted in FVC (1.21%), FEV_1_/FVC (−1.51%predicted), PEF (0.06%), and FEF_25–75_ (−2.76%). Logistic regression analysis indicated that a higher body mass index was associated with a lower risk of abnormal lung function. **Conclusions**: Ventilatory defects were observed in one-third of patients with mild COVID-19 who did not require hospitalization, mainly presenting as restrictive patterns and small airway disease. Even mild cases may have residual pulmonary impairment, warranting further long-term studies.

## 1. Introduction

Coronavirus disease 2019 (COVID-19) is a viral pandemic that has affected human health worldwide. The infection has impacted 774 million individuals globally, resulting in approximately 7 million fatalities [1]. In Thailand, multiple increasingly numerous waves of infections overwhelmed hospital capacities. Consequently, the Department of Medical Services, Ministry of Public Health, issued guidelines for managing COVID-19 patients, recommending home isolation for those with mild symptoms or symptoms improving after hospital treatment [2]. This measure aimed to reduce hospital bed occupancy rates and enhance the management of patients with severe symptoms.

The severity of COVID-19 varies from asymptomatic cases to those with upper respiratory tract infection, pneumonia, and multiorgan failure [3,4], including acute respiratory distress syndrome (ARDS) [5,6]. In COVID-19 patients with ARDS, distinct pathological features such as endothelialitis and the formation of microthrombi have been observed, alongside the development of pulmonary fibrosis [7]. Since the lungs are a critical site of infection in COVID-19, monitoring and assessing lung function during and after treatment is crucial for close management. Several studies have reported abnormal pulmonary function, ranging from 42 to 79% in recovered COVID-19 patients [8,9,10,11,12,13]. Patients with mild disease showed a decline in lung function compared to uninfected controls after up to 2 years of follow-up [14]. Patients with mild COVID-19 showed decreased forced vital capacity (FVC) and forced expiratory volume in one second (FEV_1_) in 4–9% of cases [15]. Furthermore, a reduced diffusing capacity of the lungs for carbon monoxide (DLCO) was found in 20–82% of COVID-19 patients [9,11,12,13,16,17,18,19,20,21,22], particularly in severe cases. Patients with pulmonary fibrosis typically exhibit decreased DLCO and hypoxemia [20,21,22].

However, most studies have focused on hospitalized patients, leaving a gap in understanding the lung function of those managed at home under self-isolation. Therefore, this study aimed to assess lung functions and identify factors influencing lung functions in post-recovery COVID-19 patients under home isolation.

## 2. Materials and Methods

### 2.1. Study Design and Participants

A prospective study was conducted in COVID-19 patients with asymptomatic or mild disease at Thammasat-Khukot Medical Center and Thammasat University Hospital, Thailand, between November 2021 and May 2022. Patients aged 18 years or older with COVID-19 confirmed by a positive SARS-CoV-2 reverse transcription polymerase chain reaction (RT-PCR) test, no infiltrates on chest radiographs, and no COVID-19 symptoms or mild upper respiratory symptoms at initial consultation were included. Exclusion criteria were recent myocardial infarction, blood pressure higher than 180/100 mmHg, resting heart rate greater than 120 beats per minute, and inability to perform spirometry.

Ethic approval was obtained from the Human Research Ethics Committee of Thammasat University (Medicine), Thailand (IRB No. MTU-EC-IM-0-300/64, COA No. 295/2021; date of approval: 17 November 2021), in full compliance with international guidelines, including the Declaration of Helsinki, the Belmont Report, CIOMS Guidelines, and the International Conference on Harmonization-Good Clinical Practice (ICH-GCP). All methods were performed in accordance with these guidelines and regulations. Written informed consent was obtained from all participants. This study was registered on Thaiclinicaltrials.org with the number TCTR20211121001.

### 2.2. Study Procedures

Demographic data, pre-existing comorbidities, respiratory symptoms, and lung functions by spirometry, including FVC, FEV_1_, peak expiration flow (PEF), forced expiration flow rate at 25–75% of FVC (FEF_25–75_), and bronchodilator responsiveness (BDR), were collected. Spirometry was carried out according to the international guidelines of the United States and Europe [23,24,25] using a PC-based spirometer (Vyntus SPIRO, Vyaire Medical, Mettawa, IL, USA). Briefly, participants were asked to blow into the tube hard and fast and then continue exhaling for 15 s or more. FVC, FEV_1_, FEV_1_/FVC, PEF, and FEF_25–75_ were reported in liters (L), %predicted, %, or liters per second (L/s). BDR was tested by inhaling 400 µg of salbutamol and repeating spirometry after 15 min [23,24,25]. The predicted values of all spirometry parameters were used according to the reference equations of the Global Lung Function Initiative [26]. Spirometry was performed after disease resolution at baseline and 3-month follow-up. The resolution of infection was defined as the resolution of respiratory symptoms and the second negative SARS-CoV-2 RT-PCR or rapid antigen test results.

### 2.3. Pulmonary Function Outcomes

Abnormal lung functions were assessed using specific criteria: airway obstruction defined as FEV_1_/FVC ratio < lower limit of normal (LLN); restrictive defect defined as FEV_1_/FVC > LLN and FVC < LLN [27]; mixed obstructive and restrictive defect defined as FEV_1_/FVC ratio < LLN and FVC < LLN; and small airway disease defined as FEF_25–75_ < 65% [28]. BDR was defined as FEV_1_ or FVC improvement after a BDR test of >10% of the predicted value [27]. Abnormal lung functions were reported at baseline data collection.

### 2.4. Statistical Analysis

In a previous study [22], the prevalence of restrictive defect in COVID-19 patients was 15.0%. We hypothesized that the prevalence in our population was the same. Our sample size was calculated to estimate a proportion with a power confidence of 80%, a type I error of 5%, and a precision margin of 5%. Therefore, the calculated sample size was 196.

Categorical variables were expressed as numbers (percentages), while continuous variables were expressed as mean ± standard deviation. The chi-squared test was used to compare categorical data between the abnormal and normal lung function groups, and Student’s *t*-test was used to compare the means of continuous data between the two groups. We assessed the mean changes in each variable between baseline and the 3-month follow-up. To determine the set of variables associated with abnormal lung function, we used a logistic regression model with abnormal lung function as the dependent variable. Independent variables, including age, body mass index, preexisting comorbidities, and clinical symptoms during COVID-19, were entered into the regression model if bivariate analysis indicated a statistical significance. Adjusted odds ratios (95%confidence interval) were reported for variables in the model. A two-sided *p*-value < 0.05 was considered statistically significant. Statistical analyses were performed using SPSS version 26.0 software (IBM Corp., Armonk, NY, USA).

## 3. Results

### 3.1. Participants

A total of 278 patients were screened. Of these, 250 were included in the study (58.4% female) and 28 were excluded (Figure 1). The mean age was 37.4 ± 15.2 years. Current or former smokers comprised 20.0% with an average of 6.4 ± 8.9 pack-years. Hypertension (9.6%) and hyperlipidemia (7.2%) were common preexisting comorbidities. Asthma and chronic obstructive pulmonary disease (COPD) were found in 2.0 and 0.4%, respectively. Common symptoms were fever (55.6%) and cough (60.4%) (Table 1). There were no patients with initially mild symptoms who subsequently experienced a worsening of clinical status.

### 3.2. Pulmonary Function Results

Abnormal pulmonary functions were found in 28.4% of patients (Table 1). Restrictive defect, airway obstruction, mixed obstructive and restrictive defect, and small airway disease were found in 14.4%, 5.2%, 0.4%, and 8.4%, respectively (Table 2). BDR was found in seven (2.8%) patients (Table 2); three had airway obstruction, two had restrictive defect, one had a mixed pattern, and one had small airway disease.

There were significant changes at the 3-month follow-up from baseline in FVC (1.21%, *p* = 0.015), FEV_1_/FVC (−1.51%predicted, *p* < 0.001), PEF (0.06%, *p* = 0.007), and FEF_25–75_ (−2.76%, *p* = 0.005) (Table 3).

Compared to patients with normal pulmonary functions, the abnormal pulmonary function group was significantly older, with higher proportions of hypertension, hyperlipidemia, and diabetes but a lower body mass index and lower proportions of cough, sore throat, and sneezing (Table 1). Logistic regression analysis indicated that a higher body mass index was associated with a lower risk of abnormal lung function (Table 4).

## 4. Discussion

To the best of our knowledge, this is the first study of pulmonary functions in COVID-19 patients under a home isolation program within a general population. Abnormal lung function was identified in 28.4% of patients, with the most prevalent abnormality being a restrictive defect (14.4%). However, lung restriction had improved significantly after the 3-month follow-up. A higher body mass index correlated with a lower risk of abnormal lung function. However, 5.4% of eligible patients in our cohort—who were generally average and had no severe comorbidities—were unable to perform spirometry. This may be attributed to the short duration of the spirometry test (less than 15 s) or post-COVID-19 coughing during the procedure. These factors likely contributed to the failure to meet the acceptability criteria for spirometry interpretation.

Asthma and COPD were found in 2.0 and 0.4% of participants in our study, respectively. These prevalence rates are lower than those reported in a recent asthma and COPD survey in Thailand by Saiphoklang N et al. [29], which found rates of 10.3% and 8.3%, respectively. The lower prevalence in our study may be due to selection bias, as indicated by the younger average age of participants (mean age of 37 years) compared to the mean age of 56 years in the study of Saiphoklang N. The younger age of our participants may be explained by the criteria for the home isolation program of the Ministry of Public Health of Thailand, which specified that asymptomatic or mild COVID-19 patients aged less than 60 years without comorbidities were eligible to participate in the home isolation program [30].

Our study found that hypertension, hyperlipidemia, and diabetes were associated with abnormal lung function in the univariable analysis. However, these associations may support an age-related hypothesis, suggesting that some conditions were likely preexisting and not necessarily caused by COVID-19. This is further supported by the logistic regression analysis, which showed no significant association between these variables and abnormal lung function. Moreover, our study found a statistically significant improvement in FVC at the 3-month follow-up (an increase of 1.2% in predicted value); however, this change may not be clinically meaningful.

Approximately 80% of COVID-19 patients present mild symptoms, and most of these individuals should be able to recover at home [31]. The practice of home isolation, particularly in patients who are relatively young and have no underlying medical conditions, benefits clinical management [32]. Furthermore, it aids recovery from depression and anxiety better than general quarantine [33] and helps alleviate stress on medical personnel [34].

The home isolation in our study was similar to the self-isolation of 24 athletes with COVID-19 in a study by Komici K and colleagues in Italy [35]. They found that peak oxygen uptake on cardiopulmonary exercise testing and various spirometry measurements did not differ significantly between COVID-19 patients and those who were healthy, except for FEV_1_%, which was significantly lower in the COVID-19 group. A prospective study in Denmark by Iversen KK et al. [14] found that patients with mild COVID-19 had a reduction in FEV_1_ of 13.0 mL per year from before infection to 6 months after infection compared to uninfected controls. From 6 to 24 months after infection, they experienced an additional decline of 7.5 mL per year. A similar pattern was observed for FVC. In contrast, a study in Norway by Lund Berven L et al. [36] found that non-hospitalized adolescents and young adults with mild COVID-19 showed no significant differences in spirometry parameters, including FEV_1_ and FVC, compared to non-COVID controls after a total follow-up period of 12 months.

Although most COVID-19 patients under self-isolation or home isolation experience only mild symptoms, some develop severe disease and life-threatening conditions. A study by Suess C et al. investigated morphological findings in a COVID-19 patient who died during self-isolation. They discovered massive bilateral alveolar damage, indicating early-phase ARDS [37].

The most common abnormal lung function in hospitalized COVID-19 patients is a restrictive defect (15–54%) [8,21,22,38,39], which aligns with our results. However, it is important to note that the patients in our study were under home isolation and had normal chest radiographs.

A study in Malaysia by Chai CS et al. [40] found that 46.8% of post-discharge patients recovering from moderate-to-critical COVID-19 had abnormal lung function results: 28.4% exhibited a restrictive pattern, 17.4% showed preserved ratio impaired spirometry (PRISm), and 1.0% displayed an obstructive pattern. In a study by Ekbom E. and colleagues [12] in Sweden, 52% of COVID-19 patients who recovered from an intensive care unit (ICU) after approximately 3–6 months exhibited abnormal lung function. Among them, 45% had impaired gas exchange (decreased DLCO), and 13% had impaired FVC values. However, a study by Lerum TV and colleagues in Norway [19] found that only 24% of patients had DLCO impairment at 3 months after recovery, and there were no significant differences in spirometry data, 6 min walking distance, and DLCO between ICU patients and non-ICU patients. A study in the Czech Republic by Genzor S et al. [41] found that patients with COVID-19 had decreased DLCO values—86.3%, 79%, and 68% of the predicted values for mild, moderate, and severe COVID-19, respectively.

The prevalence of abnormal spirometry in our study, 28.4%, was higher than that in a study conducted in Thailand by Eksombatchai D and colleagues [8]. They found that 17.2% of hospitalized patients had abnormal spirometry at 2 months after recovering from COVID-19 pneumonia. In the severe pneumonia group, the prevalence of abnormal spirometry was significantly higher compared to the mild and non-severe groups. Additionally, patients with abnormal chest radiography findings had significantly lower FVC, FEV_1_, FEV_1_/FVC, and FEF_25–75_ compared to those with normal chest radiographs. Our patients’ higher prevalence of abnormal spirometry might result from the presence of silent abnormal lung lesions under-detected by chest radiographs.

Our study found that FVC had improved significantly at the 3-month follow-up, possibly because PEF had also improved. PEF values are determined by lung volume, airway caliber, lung elastic recoil, and expiratory muscle strength [42]. These respiratory functions may recover after SARS-CoV-2 infection for up to 3 months. However, our findings demonstrated small airway disease in 8.4% of patients and a significant decrease in FEF_25–75_ at 3 months. These findings suggest that asymptomatic or mild COVID-19 patients might develop obstructive airway diseases such as asthma or COPD in the future. The small airways are the major sites of airflow obstruction, especially in susceptible smokers, leading to an accelerated loss of lung function in the early stages of COPD [43]. Therefore, mild COVID-19 patients would benefit from long-term monitoring by spirometry and clinical assessment.

There are limitations to this study. Firstly, we did not measure other lung function tests such as lung volume, DLCO, and impulse oscillometry (IOS). Moreover, there was a high percentage of rather young and otherwise healthy adults (5.4%) who were not able to perform spirometry. Therefore, the prevalence of abnormalities in our study might be either underestimated or overestimated—for example, a restrictive defect might be due to air trapping or other respiratory conditions. Secondly, we did not have a control group (healthy subjects), so we could not confidently confirm abnormalities in lung function resulting from COVID-19. Thirdly, we did not have spirometry data before COVID-19 to compare with post-infection results, so some lung function abnormalities in our patients might have been present before COVID-19. Lastly, long-term follow-up was not conducted. Therefore, we cannot predict changes in symptoms and lung function in the future. Longer prospective studies are required to evaluate lung function and long-term clinical outcomes in asymptomatic or mild COVID-19 patients.

## 5. Conclusions

In this prospective study of patients with mild or asymptomatic COVID-19 managed under a home isolation program, abnormal lung function was observed in approximately one-third of participants, with restrictive patterns and small airway disease being the most common findings. These results suggest that even mild COVID-19 cases may experience residual pulmonary function impairment after recovery. Further longitudinal studies are warranted to determine the clinical significance and long-term outcomes of these abnormalities.

## Figures and Tables

**Figure 1 medsci-13-00088-f001:**
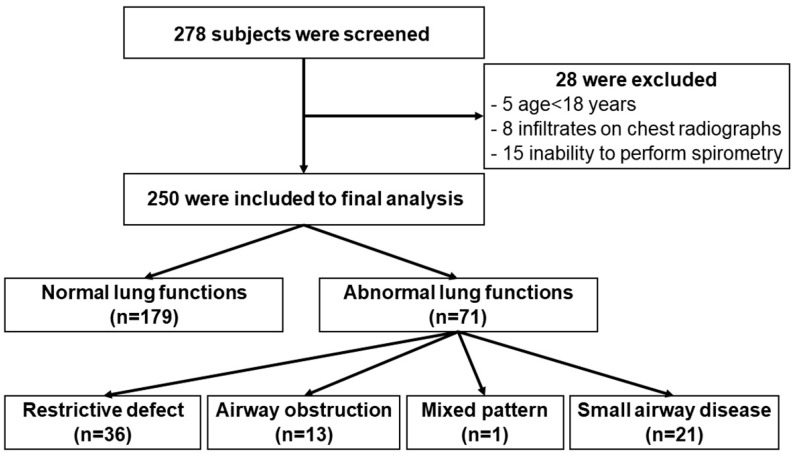
Flowchart of participant recruitment to the study.

**Table 1 medsci-13-00088-t001:** Baseline characteristics of COVID-19 patients under a home isolation program.

Characteristics	Total (n = 250)	Normal Lung Function (n = 179)	Abnormal Lung Function (n = 71)	*p*-Value
Age, years	37.4 ± 15.2	36.0 ± 14.5	41.0 ± 16.6	0.031
Female	146 (58.4)	105 (58.7)	41 (57.7)	0.895
Male	104 (41.6)	74 (41.3)	30 (42.3)	0.895
Body mass index, kg/m^2^	23.8 ± 4.9	24.2 ± 4.8	22.8 ± 5.1	0.042
Smoking	50 (20)	33 (18.4)	17 (23.9)	0.165
Amount of smoking, pack-years	6.4 ± 8.9	4.4 ± 6.5	10.3 ± 11.7	0.130
**Preexisting comorbidities**				
Hypertension	24 (9.6)	11 (6.1)	13 (18.3)	0.003
Hyperlipidemia	18 (7.2)	8 (4.5)	10 (14.1)	0.008
Diabetes	13 (5.2)	5 (2.8)	8 (11.3)	0.011
Coronary heart disease	2 (0.8)	1 (0.6)	1 (1.4)	0.488
Cerebrovascular disease	4 (1.6)	2 (1.1)	2 (2.8)	0.320
Obesity	2 (0.8)	2 (1.1)	0 (0)	1.000
Allergic rhinitis	44 (17.6)	35 (19.6)	9 (12.7)	0.198
Asthma	5 (2.0)	2 (1.1)	3 (4.2)	0.140
COPD	1 (0.4)	0 (0)	1 (1.4)	1.000
**Symptoms during COVID-19**				
No symptom	19 (7.6)	14 (7.8)	5 (7.0)	0.834
Fever	139 (55.6)	100 (55.9)	39 (54.9)	0.893
Cough	151 (60.4)	117 (65.4)	34 (47.9)	0.011
Breathlessness	79 (31.6)	57 (31.8)	22 (31.0)	0.895
Muscle pain	80 (32.0)	61 (34.1)	19 (26.8)	0.263
Headache	68 (27.2)	52 (29.1)	16 (22.5)	0.297
Sore throat	125 (50.0)	99 (55.3)	26 (36.6)	0.008
Chest tightness	13 (5.2)	8 (4.5)	5 (7.0)	0.527
Diarrhea	28 (11.2)	23 (12.8)	5 (7.0)	0.189
Vomiting	5 (2.0)	4 (2.2)	1 91.4)	1.000
Nasal obstruction	87 (34.8)	63 (35.2)	24 (33.8)	0.835
Runny nose	99 (36.6)	76 (42.5)	23 (32.4)	0.142
Sneezing	57 (22.8)	48 (26.8)	9 (12.7)	0.016
Anosmia	89 (35.6)	61 (34.1)	28 (39.4)	0.425
Ageusia	63 (25.2)	48 (26.8)	15 (21.1)	0.350

Data shown as n (%) or mean ± SD. COPD = Chronic obstructive pulmonary disease; COVID-19 = coronavirus disease 2019; kg = kilogram; m = meter.

**Table 2 medsci-13-00088-t002:** Abnormal pulmonary functions of COVID-19 patients under the home isolation program.

Abnormality	Data (n = 250)
Restrictive defect	36 (14.4)
Airway obstruction	13 (5.2)
Mixed obstructive and restrictive defect	1 (0.4)
Small airway disease	21 (8.4)
Bronchodilator response	7 (2.8)

Data shown as n (%). Airway obstruction defined as FEV_1_/FVC ratio < lower limit of normal (LLN). Restrictive defect defined as FEV_1_/FVC ratio > LLN and FVC < LLN. Mixed obstructive and restrictive defect defined as FEV_1_/FVC ratio < LLN and FVC < LLN. Small airway disease defined as FEF_25–75_ < 65% while normal FEV_1_, FVC, and FEV_1_/FVC ratio; BDR defined as an increase in FEV_1_ or FVC for >10% of the predicted value after BDR test. BDR = Bronchodilator response; FEV_1_ = forced expiratory volume in one second; FVC = forced vital capacity; FEF_25–75_ = forced expiratory flow at 25–75% of FVC.

**Table 3 medsci-13-00088-t003:** Pulmonary function data of COVID-19 patients at baseline and 3-month follow-up.

Parameters	Baseline (n = 250)	3-Month Follow-Up (n = 200)	Mean Change (95% CI)	*p*-Value
FVC, L	3.24 ± 0.84	3.24 ± 0.86	−0.005 (−0.021, 0.031)	0.717
FVC, %predicted	94.19 ± 13.91	95.41 ± 14.81	1.217 (0.237, 2.197)	0.015
FEV_1_, L	2.73 ± 0.74	2.72 ± 0.77	−0.009 (−0.032, 0.013)	0.407
FEV_1_, % predicted	94.16 ± 14.66	94.75 ± 14.99	0.593 (−0.320, 1.507)	0.201
FEV_1_ change after BD test, %	2.50 ± 3.44	2.44 ± 3.29	−0.057 (−0.677, 0.563)	0.856
FVC change after BD test, %	0.28 ± 3.33	−0.30 ± 3.24	−0.580 (−1.253, 0.093)	0.091
FEV_1_/FVC, %	84.50 ± 8.23	84.26 ± 7.96	−0.237 (−0.760, 0.285)	0.370
FEV_1_/FVC, % predicted	104.69 ± 9.27	103.18 ± 8.98	−1.511 (−2.201, −0.821)	<0.001
PEF, L/s	6.97 ± 1.90	7.17 ± 1.95	0.193 (0.064, 0.323)	0.004
PEF, % predicted	94.03 ± 17.38	96.39 ± 17.48	0.064 (0.642, 4.082)	0.007
FEF_25–75_, L/s	3.09 ± 1.28	3.01 ± 1.31	−0.080 (−0.144, −0.016)	0.015
FEF_25–75_, %predicted	88.18 ± 25.83	85.42 ± 26.41	−2.762 (−4.682, −0.843)	0.005

Data shown as mean ± SD. BD = Bronchodilator response; FEV_1_ = forced expiratory volume in 1 s; FVC = forced vital capacity; FEF_25–75_ = forced expiratory flow at 25–75% of FVC; PEF = peak expiratory flow; L = liters; s = second.

**Table 4 medsci-13-00088-t004:** Logistic regression analysis for factors associated with abnormal pulmonary functions.

Variables	Adjusted Odds Ratio (95%CI)	*p*-Value
Age for every 1-year increase	0.999 (0.976–1.024)	0.967
Body mass index for every 1-unit increase	0.898 (0.836–0.965)	0.004
Hypertension	0.458 (0.136–1.539)	0.207
Hyperlipidemia	0.352 (0.099–1.248)	0.106
Diabetes	0.394 (0.088–1.766)	0.224
Cough	1.517 (0.787–2.923)	0.213
Sore throat	1.441 (0.723–2.874)	0.299
Sneezing	2.127 (0.905–5.000)	0.083

## Data Availability

The data supporting the results of this study are available within the article.

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
