# Peer review of "Pulmonary Function Among COVID-19 Patients in Home Isolation Program"

_medsci, 2025, doi:10.3390/medsci13030088_

Round 1
Reviewer 1 Report
Comments and Suggestions for Authors
Dear colleagues
Thank you for your effort in evaluation pulmonary function testing in post-COVID19 patients with mild disease.
There are some limitations of your paper:
- 5.4% of eligible patients were unable to perform spirometry in an adult cohort of rather average patients with no severe comorbid diseases ["Exclusion criteria were recent myocardial infarction, blood pressure higher than 180/100 mmHg, resting heart rate greater than 120 beats per minute]. This percentage is rather unusual and need some clarification statements. Could it be related to duration of test? In rows 92-94 you are stating "Briefly, participants were asked to blow into the tube hard and fast and then to continue exhaling for 15 seconds or more."
- In rows 132-133 you are stating "Asthma and chronic obstructive pulmonary disease (COPD) were found in 2.0 and 0.4%, respectively." How does these numbers correlate with national data from your country or with previous local studies? Numbers seem to be rather small and they could be generated by a selection bias.
- Issue in precedent comment could have been avoided by including a control group of non-COVID patients tested in same approach. The number of included patients was so small so testing of a control group could have been feasible, as you are stating in limitations section (rows 223-224).
- In Discussion section rows 168-169 you are stating "To the best of your knowledge, this is the first study of pulmonary functions in COVID-19 patients under a home isolation program within a general population." So comparing findings in a general population sample with specific subgroups is not accurate. Comparing your findings with Komici's paper [reference 33] that has evaluated 24 athletes with COVID-19 is not a reasonable approach. Same comment about comparison with a subgroup of severe patients recovered from ICU - Ekbom's paper (reference 12).
- Conclusion section must be revisited because you are speculating future outcomes without any proof in these presented data.
Author Response
General comment: Thank you for your effort in evaluation pulmonary function testing in post-COVID19 patients with mild disease.
Response: We would like to express our heartfelt gratitude to the reviewer for the wonderful reviews and comments. We will do our best to address all suggestions accordingly.
Comments 1: 5.4% of eligible patients were unable to perform spirometry in an adult cohort of rather average patients with no severe comorbid diseases ["Exclusion criteria were recent myocardial infarction, blood pressure higher than 180/100 mmHg, resting heart rate greater than 120 beats per minute]. This percentage is rather unusual and need some clarification statements. Could it be related to duration of test? In rows 92-94 you are stating "Briefly, participants were asked to blow into the tube hard and fast and then to continue exhaling for 15 seconds or more.
Response 1: We have addressed this issue in the Discussion section (page 6, lines 177–181). Thank you very much.
Comments 2: In rows 132-133 you are stating "Asthma and chronic obstructive pulmonary disease (COPD) were found in 2.0 and 0.4%, respectively." How does these numbers correlate with national data from your country or with previous local studies? Numbers seem to be rather small and they could be generated by a selection bias.
Response 2: We have discussed this point in the Discussion section (page 6, lines 182–187). Thank you for your valuable feedback.
Comments 3: Issue in precedent comment could have been avoided by including a control group of non-COVID patients tested in same approach. The number of included patients was so small so testing of a control group could have been feasible, as you are stating in limitations section (rows 223-224).
Response 3: This issue regarding the control group is a point of concern for us. We have addressed this in the Discussion section (page 8, lines 261–262). Thank you very much.
Comments 4: In Discussion section rows 168-169 you are stating "To the best of your knowledge, this is the first study of pulmonary functions in COVID-19 patients under a home isolation program within a general population." So comparing findings in a general population sample with specific subgroups is not accurate. Comparing your findings with Komici's paper [reference 33] that has evaluated 24 athletes with COVID-19 is not a reasonable approach. Same comment about comparison with a subgroup of severe patients recovered from ICU - Ekbom's paper (reference 12).
Response 4: According to an extensive review of the literature, there is no study on pulmonary function in mild COVID-19 cases with a home isolation program. However, we have added data from studies on moderate to severe COVID-19 in the Discussion section (page 7, lines 210–218). Thank you very much.
Comments 5: Conclusion section must be revisited because you are speculating future outcomes without any proof in these presented data.
Response 5: We have revised the text in the Conclusion section (page 1, lines 36–39 and page 8, lines 270–276). Thank you very much.

Reviewer 2 Report
Comments and Suggestions for Authors
The article is focused on the ventilatory defects potentially linked to a mild forms of SarsCov2 infection – potentially interesting since different opinions exist.
Introduction contains generic data concerning SarsCov2 infection; perhaps more information on lung function impact would be useful to create a framework for the article.
Material and methods
The study procedures are adequately described.
The practice of including factors in a logistic regression based on bivariate comparisons on the same dataset is not robust – this should be checked by a statistics expert.
Results are clearly presented and according to the variables presented in the material and methods section.
Multiple comparisons seem to have been performed on the same data set therefore some form of correction should have been implemented when assessing the p value – see table 1.
Discussion seem underdeveloped -since there is ample literature relevant to lung function impairment following COVID. Furthermore, persistent ventilatory defects after three months should be considered long COVID further expanding the discussion.
Some issues should have been considered
- If the lung function anomalies are necessarily linked to the COVID episode – the connection with hypertension, dislipidemia and diabetes might support an age hypothesis (perhaps at least some were preexistent and not necessary caused by the SarsCov2 infection)
- Diagnosing a restrictive defect using spirometry alone is not accurate – as the same pattern might be caused by air trapping or other respiratory conditions
- Statistical improvement is somewhat debatable in clinical terms since both values are within the normal range
‘Our study found that FVC had improved significantly at the 3-month follow-up’
|
FVC, L |
3.24±0.84 |
3.24±0.86 |
-0.005 (-0.021, 0.031) |
0.717 |
|
FVC, %predicted |
94.19±13.91 |
95.41±14.81 |
1.217 (0.237, 2.197) |
0.015 |
Conclusions
‘Abnormal pulmonary functions, particularly a restrictive lung pattern, were common in asymptomatic or mild COVID-19 patients under a home isolation program’ should be rephrased as the results do not support the use of ‘common’
‘However, other findings suggest that mild COVID-19 patients might develop airway obstruction due to small airway disease in the future.’ should probably be removed since it is not related to the study.
Perhaps ‘ventilatory defects were present in mild forms of COVID patients not requiring hospitalization..’ captures better the essence of the article.
Comments on the Quality of English LanguageSome rephrasing might be needed.
Author Response
General comment: The article is focused on the ventilatory defects potentially linked to a mild form of SarsCov2 infection – potentially interesting since different opinions exist.
Response: We would like to express our heartfelt gratitude to the reviewer for the wonderful reviews and comments. We will do our best to address all suggestions accordingly.
Comments 1: Introduction contains generic data concerning SarsCov2 infection; perhaps more information on lung function impact would be useful to create a framework for the article.
Response 1: Thank you for your valuable suggestion. We have added more information to the Introduction section (page 2, lines 59–62).
Comments 2: Material and methods
The study procedures are adequately described.
The practice of including factors in a logistic regression based on bivariate comparisons on the same dataset is not robust – this should be checked by a statistics expert.
Response 2: Thank you for your insightful comment. Our statistical analysis has been checked by a statistics expert.
Comments 3: Results are clearly presented and according to the variables presented in the material and methods section.
Response 3: Thank you for your valuable review and suggestions.
Comments 4: Multiple comparisons seem to have been performed on the same data set therefore some form of correction should have been implemented when assessing the p value – see table 1.
Response 4: We have reassessed the categorization of all variables and the corresponding p-values, and confirm that all are correct. Therefore, we present the current data without any correction.
Comments 5: Discussion seem underdeveloped -since there is ample literature relevant to lung function impairment following COVID. Furthermore, persistent ventilatory defects after three months should be considered long COVID further expanding the discussion.
Response 5: We have included relevant studies on post-COVID lung function, including long-term sequelae, in the Discussion section (page 7, lines 210–218, 224–227, and 234–236). Thank you very much.
Comments 6: Some issues should have been considered.
If the lung function anomalies are necessarily linked to the COVID episode – the connection with hypertension, dislipidemia and diabetes might support an age hypothesis (perhaps at least some were preexistent and not necessary caused by the SarsCov2 infection).
Response 6: We have addressed this point in the Discussion section (page 7, lines 188–193).
Comments 7: Diagnosing a restrictive defect using spirometry alone is not accurate – as the same pattern might be caused by air trapping or other respiratory conditions.
Response 7: We have addressed this point in the Limitations section (page 8, lines 257–261).
Comments 8: Statistical improvement is somewhat debatable in clinical terms since both values are within the normal range.
‘Our study found that FVC had improved significantly at the 3-month follow-up’
|
VC, L |
3.24±0.84 |
3.24±0.86 |
-0.005 (-0.021, 0.031) |
0.717 |
|
FVC, %predicted |
94.19±13.91 |
95.41±14.81 |
1.217 (0.237, 2.197) |
0.015 |
Response 8: We have addressed this point in the Discussion section (page 7, lines 193–195).
Comments 9: Conclusions
‘Abnormal pulmonary functions, particularly a restrictive lung pattern, were common in asymptomatic or mild COVID-19 patients under a home isolation program’ should be rephrased as the results do not support the use of ‘common’.
Response 9: We have revised the text in the Conclusion section (page 1, lines 36–39 and page 8, lines 270–276). Thank you very much.
Comments 10: ‘However, other findings suggest that mild COVID-19 patients might develop airway obstruction due to small airway disease in the future.’ should probably be removed since it is not related to the study.
Response 10: We have removed this sentence from the Conclusion section (page 1, lines 36–39 and page 8, lines 270–276).
Comments 11: Perhaps ‘ventilatory defects were present in mild forms of COVID patients not requiring hospitalization.’ captures better the essence of the article.
Response 11: We have added this point to the Conclusion section (page 1, lines 36–39).
Comments 12: Some rephrasing might be needed.
Response 12: We have rephrased it according to your suggestions. Thank you very much.

Reviewer 3 Report
Comments and Suggestions for Authors
I would advise changing the title to:
'Pulmonary Function among COVID-19 Patients in a Home Isolation Program'.
Please can the authors address the following issues:
- You cannot confirm lung restriction without doing volumes - this needs to be explicitly stated.
- Why did the authors not look as gas transfer when COVID-19 is primarily a disease affecting lung parenchyma?
- Why weren't patients with existing lung abnormalities excluded?
- Why wasn't smoking looked at in table 4?
- I think it is difficult to claim that patients with small airways disease in this context will go on to have obstructive lung disease. Where is the evidence for this?
This was generally ok with one or two examples that could be improved such as the title.
Author Response
General comment: I would advise changing the title to:
'Pulmonary Function among COVID-19 Patients in a Home Isolation Program'.
Response: We have revised the title as per your suggestion (page 1, line 2). Thank you very much.
Comments 1: Please can the authors address the following issues:
You cannot confirm lung restriction without doing volumes - this needs to be explicitly stated.
Response 1: We have addressed this point in the Limitations section (page 8, lines 257–261).
Comments 2: Why did the authors not look as gas transfer when COVID-19 is primarily a disease affecting lung parenchyma?
Response 2: Thank you for your insightful comment. We acknowledge that this measurement is a limitation of our study. We have addressed this point in the Limitations section (page 8, lines 257–261).
Comments 3: Why weren't patients with existing lung abnormalities excluded?
Response 3: Our aim was to investigate the progression of pulmonary function after COVID-19 in patients both with and without pre-existing lung disease. Therefore, we included patients with existing lung abnormalities in our study. Thank you for the suggestion.
Comments 4: Why wasn't smoking looked at in table 4?
Response 4: We included only the significant variables from the bivariable analysis in Table 1 in the multivariable (logistic regression) analysis presented in Table 4. Therefore, smoking was not included in the logistic regression analysis.
Comments 5: I think it is difficult to claim that patients with small airways disease in this context will go on to have obstructive lung disease. Where is the evidence for this?
Response 5: We have removed this point from the Conclusion section (page 1, lines 36–39 and page 8, lines 270–276). Thank you very much.
Comments 6: This was generally ok with one or two examples that could be improved such as the title.
Response 6: We would like to express our heartfelt gratitude to the reviewer for the wonderful reviews and comments.

Round 2
Reviewer 1 Report
Comments and Suggestions for Authors
Dear colleagues thank you for your efforts to improve your paper.
Some serious issues are still present.
Methodology of study is not clearly stated: patients were COVID-19 patients confirmed in the 2 hospitals that were evaluated as mild at initial consultation and were screened at that moment or were selected from a convalescent cohort of patients that was previously managed as outpatients? Because compliance to treatment protocol and timely presentation are very different in such cohorts and could influence outcomes.
There are no data of patients that were initially mild and had subsequent worsening of clinical status nor data on time between diagnosis and inclusion in study. It is not plausible that all patients labelled as mild disease would have remained as such, with no worsening of disease, especially at that moment of pandemic (during Omicron wave from November 2021 to May 2022).
Seems odd the criterion of having a second negative COVID-19 test for being eligible for this study... why did you select convalescent patients only after a second negative test? What is the clinical relevance of this request for inclusion?
Seems strange the high percentage of rather young and otherwise healthy adults that were not able to perform spirometry. During informed-consent discussion potential study subjects were informed that study includes pulmonary function testing. So it seems strange that such a significant number of people believed that they will be able to perform PFT and finally to fail procedure in spite of young age - 37 years on average, as you calculated.
Selection bias is evident when you compare asthma and COPD in your cohort with previously published data from your country [reference 29]. Asthma was 5 times less frequent and COPD 20 times less frequent than in general population so study group is not representative for general population of Thailand and findings could not be extrapolated to future epidemic outbreaks.
Discussion section is includes a confusing variability of papers that are not presenting comparable cohorts with present study. You are, moving from severe COVID cohorts [references 8,21,22,30,31 are related to hospitalized patients] to mild disease cohorts [references 14,32,37 are related to non-hospitalized patients], ICU patients [references 12 and are related to moderate to critical hospitalized patients] to athletes in self-isolation [reference 36] and so on.
You have to order these papers and to generate meaningful similitude elements
Author Response
General comment: Dear colleagues thank you for your efforts to improve your paper.
Some serious issues are still present.
Response: We would like to express our heartfelt gratitude to the reviewer for the wonderful reviews and comments. We will do our best to address all suggestions accordingly.
Comments 1: Methodology of study is not clearly stated: patients were COVID-19 patients confirmed in the 2 hospitals that were evaluated as mild at initial consultation and were screened at that moment or were selected from a convalescent cohort of patients that was previously managed as outpatients? Because compliance to treatment protocol and timely presentation are very different in such cohorts and could influence outcomes.
Response 1: We have addressed this issue in the Study Design and Participants section (page 2, line 78). Thank you very much.
Comments 2: There are no data of patients that were initially mild and had subsequent worsening of clinical status nor data on time between diagnosis and inclusion in study. It is not plausible that all patients labelled as mild disease would have remained as such, with no worsening of disease, especially at that moment of pandemic (during Omicron wave from November 2021 to May 2022).
Response 2: We confirm that there were no patients with initially mild symptoms who subsequently experienced worsening of clinical status. We have added this result in the Results section (pages 3–4, lines 137–139). Thank you for your valuable feedback.
Comments 3: Seems odd the criterion of having a second negative COVID-19 test for being eligible for this study... why did you select convalescent patients only after a second negative test? What is the clinical relevance of this request for inclusion?
Response 3: The requirement for a second negative COVID-19 test to be eligible for this study was implemented for infection control purposes during spirometry testing, in order to prevent the potential spread of COVID-19. We assumed that patients with a second negative COVID-19 test would have a very low risk of active SARS-CoV-2 infection. Thank you very much for your comments.
Comments 4: Seems strange the high percentage of rather young and otherwise healthy adults that were not able to perform spirometry. During informed-consent discussion potential study subjects were informed that study includes pulmonary function testing. So it seems strange that such a significant number of people believed that they will be able to perform PFT and finally to fail procedure in spite of young age - 37 years on average, as you calculated.
Response 4: We have addressed this issue in the Discussion section (page 6, lines 178–182) and in the Limitations section (page 8, lines 259–261). Thank you very much.
Comments 5: Selection bias is evident when you compare asthma and COPD in your cohort with previously published data from your country [reference 29]. Asthma was 5 times less frequent and COPD 20 times less frequent than in general population so study group is not representative for general population of Thailand and findings could not be extrapolated to future epidemic outbreaks.
Response 5: We have addressed this issue and added the explanation for the younger age of participants in the Discussion section (pages 6–7, lines 184–193). Thank you very much.
Comments 6: Discussion section is includes a confusing variability of papers that are not presenting comparable cohorts with present study. You are, moving from severe COVID cohorts [references 8,21,22,30,31 are related to hospitalized patients] to mild disease cohorts [references 14,32,37 are related to non-hospitalized patients], ICU patients [references 12 and are related to moderate to critical hospitalized patients] to athletes in self-isolation [reference 36] and so on.
You have to order these papers and to generate meaningful similitude elements.
Response 6: We have reordered the discussion according to the severity of disease, from mild cases (page 7, lines 202–220) to moderate to severe hospitalized case (page 7, lines 221–242) in the Discussion section. Thank you very much for your valuable comment.
Reviewer 2 Report
Comments and Suggestions for Authors
The authors improved the manuscript.
'These findings suggest that asymptomatic or mild COVID-19 patients
might develop obstructive airway diseases such as asthma or COPD in the future. The small airways are the major sites of airflow obstruction, especially in susceptible smokers, leading to an accelerated loss of lung function in the early stages of COPD [41]. Therefore, mild COVID-19 patients should undergo long-term monitoring by spirometry and clinical assessment.' - perhaps a less clear cut phrase '...would benefit from long term..." since the development of asthma or COPD is not certain.
Author Response
General comment: The authors improved the manuscript.
Response: We would like to express our heartfelt gratitude to the reviewer for the wonderful reviews and comments. We will do our best to address all suggestions accordingly.
Comments 1: 'These findings suggest that asymptomatic or mild COVID-19 patients might develop obstructive airway diseases such as asthma or COPD in the future. The small airways are the major sites of airflow obstruction, especially in susceptible smokers, leading to an accelerated loss of lung function in the early stages of COPD [41]. Therefore, mild COVID-19 patients should undergo long-term monitoring by spirometry and clinical assessment.' - perhaps a less clear cut phrase '...would benefit from long term..." since the development of asthma or COPD is not certain.
Response 1: Thank you for your valuable suggestion. We have revised the text accordingly in the Discussion section (page 8, line 261).
Reviewer 3 Report
Comments and Suggestions for Authors
Thank you for making the alterations to the paper. I think that the paper can be published in this new form.
Author Response
General comment: Thank you for making the alterations to the paper. I think that the paper can be published in this new form.
Response: We would like to express our heartfelt gratitude to the reviewer for the wonderful reviews and comments.
Round 3
Reviewer 1 Report
Comments and Suggestions for Authors
Dear colleagues I would like to send you my sincere admiration for your continuous efforts of improving your paper.
Most of the comments and changes did a great job.
Still some unsupported statements are still present and generate contradictions
In rows 258-260 you are stating "Our patients with higher prevalence of abnormal spirometry might result from presence of silent abnormal lung lesions under-detected by chest radiographs." .
But in rows 188-197 you have presented convincing arguments for a healthier and younger cohort compared with general population in your country, as a valid argument for better results compared with Saiphoklang's paper [reference 29].
This second statement seems to be supported by evidence (you are stating in rows 138-139 "Current or former smokers comprised 20.0% with an average of 6.4±8.9 pack-years." that seems a rather small number for global reported values for general population, and are attributable to serious tobacco regulation policies in your country - https://tobaccoatlas.org/factsheets/thailand/).
Older people in your country have been smoking more than current young people. Two decades ago more than 43% of male population was smoking (https://globalactiontoendsmoking.org/research/tobacco-around-the-world/thailand/) so it has a low probability that a young cohort as yours to be less healthy than average population of Thailand.
Please rephrase statement in rows 258-260.
A potential explanation for results of paper cited as reference 8 (Eksombatchai D and colleagues) is that authors of reference 8 performed Lung Function Testing after 2 months [on average] post-COVID and your paper had first PFT's performed as 2 weeks post disease. Potential improvement in these 6 weeks should not be ruled out as cause of discrepancy.
Also in conclusion section you should mention that PFT's changes were documented even in young people with mild disease and rather low prevalence of tobacco and aerosolized particulate matter exposure, so further research is needed to understand the long term velocity of age-dependent lung function decline in such patients. Especially for older people, smoking patients or professionals with work-related lung hazards that had mild COVID-19 and did not perceive any functional impairment during initial disease.
In references section
References 1 and 2 have no date of access - please correct
Some references seem not to have pages for cited paper - please insert doi identifier if there are no pages listed in citation tool used by your group. As in references 6-13, 15, 17, 19, 21, 25, 27, 29, 31-33, 35, 36, 38-40 and 42
Author Response
General comment: Dear colleagues I would like to send you my sincere admiration for your continuous efforts of improving your paper.
Most of the comments and changes did a great job.
Still some unsupported statements are still present and generate contradictions.
Response: We would like to express our heartfelt gratitude to the reviewer for the wonderful reviews and comments. We will do our best to address all suggestions accordingly.
Comments 1: In rows 258-260 you are stating "Our patients with higher prevalence of abnormal spirometry might result from presence of silent abnormal lung lesions under-detected by chest radiographs." .
Comments 1: In rows 258-260 you are stating "Our patients with higher prevalence of abnormal spirometry might result from presence of silent abnormal lung lesions under-detected by chest radiographs." .
But in rows 188-197 you have presented convincing arguments for a healthier and younger cohort compared with general population in your country, as a valid argument for better results compared with Saiphoklang's paper [reference 29].
This second statement seems to be supported by evidence (you are stating in rows 138-139 "Current or former smokers comprised 20.0% with an average of 6.4±8.9 pack-years." that seems a rather small number for global reported values for general population, and are attributable to serious tobacco regulation policies in your country - https://tobaccoatlas.org/factsheets/thailand/).
Older people in your country have been smoking more than current young people. Two decades ago more than 43% of male population was smoking (https://globalactiontoendsmoking.org/research/tobacco-around-the-world/thailand/) so it has a low probability that a young cohort as yours to be less healthy than average population of Thailand.
Please rephrase statement in rows 258-260.
Response 1: We have revised the statement in the Discussion section (page 8, lines 250–254). Thank you very much.
Comments 2: A potential explanation for results of paper cited as reference 8 (Eksombatchai D and colleagues) is that authors of reference 8 performed Lung Function Testing after 2 months [on average] post-COVID and your paper had first PFT's performed as 2 weeks post disease. Potential improvement in these 6 weeks should not be ruled out as cause of discrepancy.
Response 2: We have revised the statement in the Discussion section (page 8, lines 250–254). Thank you very much. Thank you for your valuable feedback.
Comments 3: Also in conclusion section you should mention that PFT's changes were documented even in young people with mild disease and rather low prevalence of tobacco and aerosolized particulate matter exposure, so further research is needed to understand the long term velocity of age-dependent lung function decline in such patients. Especially for older people, smoking patients or professionals with work-related lung hazards that had mild COVID-19 and did not perceive any functional impairment during initial disease.
Response 3: We have revised the text in the Abstract section (page 1, lines 38–41) and in the Conclusions section (pages 8–9, lines 284–290). Thank you for your suggestions.
Comments 3: In references section
References 1 and 2 have no date of access - please correct
Some references seem not to have pages for cited paper - please insert doi identifier if there are no pages listed in citation tool used by your group. As in references 6-13, 15, 17, 19, 21, 25, 27, 29, 31-33, 35, 36, 38-40 and 42.
Response 3: We have revised the references in the References section (pages 10–12, lines 324–446). Thank you for your valuable feedback.
